# Independent control of the thermodynamic and kinetic properties of aptamer switches

Brandon D. Wilson [1,5], Amani A. Hariri[2,5], Ian A.P. Thompson [2], Michael Eisenstein[2,3] & H. Tom Soh [2,3,4]*

Molecular switches that change their conformation upon target binding offer powerful capabilities for biotechnology and synthetic biology. Aptamers are useful as molecular switches because they offer excellent binding properties, undergo reversible folding, and can be engineered into many nanostructures. Unfortunately, the thermodynamic and kinetic properties of the aptamer switches developed to date are intrinsically coupled, such that high temporal resolution can only be achieved at the cost of lower sensitivity or high background. Here, we describe a design strategy that decouples and enables independent control over the thermodynamics and kinetics of aptamer switches. Starting from a single aptamer, we create an array of aptamer switches with effective dissociation constants ranging from 10 µM to 40 mM and binding kinetics ranging from 170 ms to 3 s. Our strategy is broadly applicable to other aptamers, enabling the development of switches suitable for a diverse range of biotechnology applications.

[1] Department of Chemical Engineering, Stanford University, Stanford, CA 94305, USA. [2] Department of Electrical Engineering, Stanford University, Stanford, CA 94305, USA. [3] Department of Radiology, Stanford University, Stanford, CA 94305, USA. [4] Chan Zuckerberg Biohub, San Francisco, CA 94158, USA. [5] These authors contributed equally: Brandon D. Wilson, Amani A. Hariri *email: tsoh@stanford.edu

A wide range of essential biological functions are governed by the action of molecular switches[1,2], which undergo a reversible conformational change upon binding a specific target molecule. These switches can be coupled to other molecular machinery to trigger a wide range of downstream functions. There is considerable interest in engineering biologically inspired molecular switches that can achieve a selective and sensitive output in response to binding a target molecule, which could prove valuable for diverse applications, including imaging[3,4], biosensing[5], and drug delivery[6,7]. Aptamers have proven to be particularly promising and versatile in this regard[8] as they are highly stable, easy to synthesize, exhibit reversible binding, and are readily adaptable to chemical modification. Since conventional methods of aptamer generation do not routinely yield aptamers capable of structure switching, many selection schemes[9,10] and engineering approaches[11–15] have been developed for the creation of aptamer switches. In contrast to naturally occurring molecular switches that have evolved over millions of years to function under precise physiological conditions, switches based on synthetic affinity reagents must be tuned to match their intended function. Unfortunately, existing selection and engineering strategies offer limited control over the thermodynamic and kinetic properties of the resultant aptamer switches and, by extension, over properties such as effective binding affinity and temporal resolution.

Recent work has yielded important insights into how to control the binding of molecular switches. For instance, it has been shown that the hybridization strength of the hairpin in a molecular beacon can modulate the effective detection range for target concentrations spanning many orders of magnitude[16]. Recognizing that this enthalpy-driven control is coarse-grained, other work achieved fine-grained control of effective binding affinity through the modulation of the entropic change associated with the degree of confinement imposed by a intramolecular linker of variable length[17]. However, previous efforts at tuning the binding properties of aptamer constructs have found that their thermodynamic and kinetic properties are intrinsically coupled[18], such that fast temporal resolution can only be achieved at the cost of either large background signal or lower affinity, or requires high-temperature conditions that can interfere with ligand binding[19]. At present, there is no reliable strategy for independently controlling the thermodynamic and kinetic properties of engineered aptamer switches.

Here, we introduce a general framework for the design of aptamer switches that enables independent control over their thermodynamic and kinetic properties. We explore the use of an intramolecular strand-displacement (ISD) strategy and the degree to which the binding properties of this construct can be controlled through rational design. Our ISD construct consists of a single-molecule switch in which an aptamer is attached to a partially complementary displacement strand via a poly-T linker (Fig. 1). Briefly, target binding to the aptamer shifts the equilibrium towards dehybridization of the displacement strand, enabling fluorescence-based target detection through disruption of a fluorophore-quencher pair. The key feature of this design is that it offers two distinct control parameters: displacement strand length ($L_{DS}$) and loop length ($L_{loop}$). This is in contrast to alternative constructs such as aptamer beacons, which have just a single control parameter—displacement strand length—that confers only coarse-grained control over affinity and couples the construct's kinetics to its thermodynamics. We show mathematically and experimentally that the two control parameters of the ISD design enable us to precisely and independently tune the thermodynamics and kinetics of the resulting aptamer switches. We use this approach to generate an array of aptamer switches that exhibit affinities spanning four orders of magnitude, with

equilibrium dissociation constants ($K_D$) ranging from 10 μM to 40 mM and binding kinetics ranging from 170 ms to 3 s—all starting from the same parent ATP aptamer. Lastly, we demonstrate that even tighter control of binding affinity and kinetics can be achieved by introducing single-base mismatches into the displacement strand. This approach is broadly applicable to virtually any aptamer, enabling facile production of highly controllable molecular switches that respond to ligands over a wide range of concentrations and time scales.

## Results

**Design and rationale**. The ISD design achieves molecular recognition through concentration-dependent shifts in equilibrium (Fig. 1). A fluorophore and a quencher are added to the 5′- and 3′-ends of the construct, respectively, enabling a fluorescent readout of target concentration (Supplementary Fig. 1a). Although the thermodynamic and kinetic parameters associated with target binding to the native aptamer ($K_D^{apt}, k_{on}^{apt}, k_{off}^{apt}$) are fixed, we can tune the overall signaling response by altering the parameters of the hybridization/quenching reaction ($K_Q, k_{on}^{DS}, k_{off}^{DS}$).

Since hairpin hybridization strength confers coarse-grained control over binding affinity[16] and linker length confers fine-grained control of binding thermodynamics[17], the incorporation of both tuning mechanisms makes this switch design highly amenable to the fine-tuning of molecular recognition (Fig. 2a). Increasing the hybridization strength of the displacement strand shifts the equilibrium towards the quenched state, which will decrease the background signal at the expense of decreased effective affinity ($\uparrow K_D^{eff}$) and temporal resolution (Fig. 2b, c). Decreasing $L_{loop}$ results in a similar equilibrium shift due to increased effective concentration of the displacement strand, but also results in increased temporal resolution. Independent tuning of the ISD's thermodynamics and kinetics is made possible by the orthogonal effects of these two parameters.

We first used a model system to mathematically test the anticipated effects on binding response, and then confirmed these effects with experimental results from an array of ISD switches (Supplementary Fig. 2, Supplementary Table 1) based on a well-studied ATP aptamer[20,21] with varying $L_{loop}$ and $L_{DS}$. All of the resulting constructs retain the high selectivity of the native aptamer (Supplementary Fig. 1b). We show conclusively that modulating $L_{loop}$ and $L_{DS}$ in tandem decouples the control over the thermodynamics and kinetics of molecular recognition. Lastly, we introduce targeted mismatches as a third tuning parameter to obtain even more precise enthalpic tuning and ultra-fast kinetics over a wide range of binding affinities.

**Principles of ISD molecular switch design**. By examining a three-state population shift model, we gain general insights into how the design parameters affect the overall thermodynamics and kinetics of molecular recognition. We have used an induced-fit model for its generalizability and simplicity, but we have also provided derivations for two-site induced-fit and conformational selection (Supplementary Figs 3 and 4, respectively). This inclusion is in recognition of the fact that the ATP aptamer used in this work has two binding sites[20] and can exhibit both induced-fit and conformational selection behavior[22–24].

We assume the switch exists in an equilibrium between quenched ($Q$), unfolded ($U$), and target-bound ($B$) states (Fig. 1). The distributions of $Q$, $U$, and $B$ are governed by the equilibrium constant ($K_Q$) for the intramolecular quenching reaction,

$$K_Q = \frac{[Q]}{[U]} = \frac{k_{on}^{DS}}{k_{off}^{DS}} \quad (1)$$

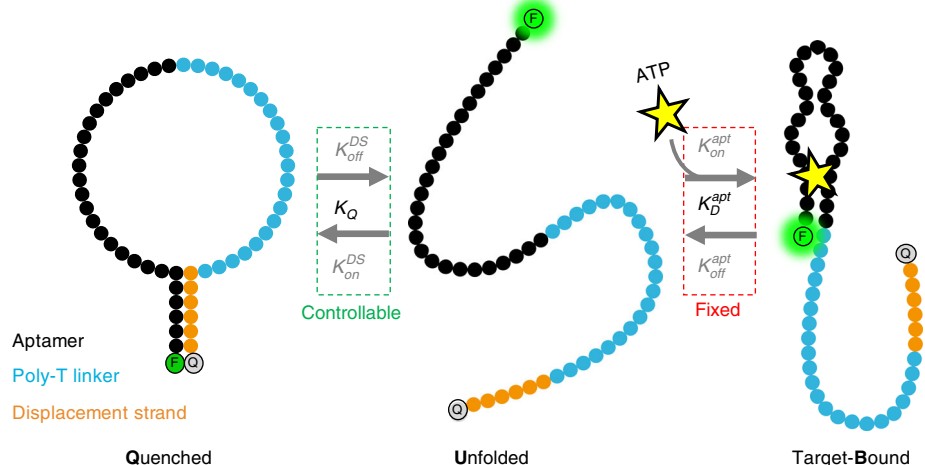

**Fig. 1** Overview of the intramolecular strand-displacement (ISD) switch design. We use an ISD design to convert an existing aptamer into a switch that gives a target-concentration-dependent signal based on the interaction of a fluorophore-quencher pair at the 5′- and 3′- ends of the construct, respectively. In this three-state population shift model, signal is generated by the unfolded and target-bound forms. In the absence of target, the quenched and unfolded states are in equilibrium, defined by $K_Q$. Target binding depletes the unfolded population, and the reaction shifts to the right, generating a signal that is proportional to target concentration. We use this model here to generalize the discussion and insights to other aptamers; however, since the ATP aptamer has two binding sites[20], we have used a modified two-site binding model for fitting and calculations (Supplementary Fig. 2, Supplementary Eq. 1)

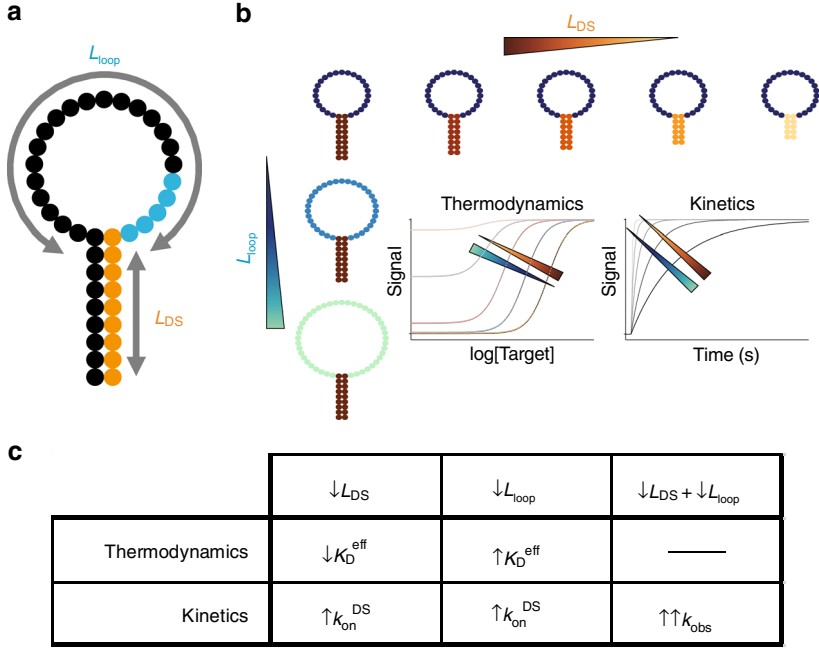

**Fig. 2** ISD control parameters and their effects. **a** By modulating the length of the linker ($L_{loop}$) and the number of bases in the hairpin ($L_{DS}$) between the displacement strand and the aptamer, we can control both the kinetics and thermodynamics of our ISD switch. We describe our switches using the nomenclature of $L_{DS}$_$L_{loop}$, i.e., a construct with a 9 bp displacement strand an a 33 nt loop is referred to as 9_33. **b** On one hand, reducing $L_{DS}$ increases effective binding affinity at the expense of increased background signal. On the other hand, decreasing $L_{loop}$ decreases background at the expense of lower effective binding affinity. **c** Decreasing either parameter increases the overall rate of binding, making it possible to increase the kinetics of a given construct while retaining the same $K_D^{eff}$

and the dissociation constant of the native aptamer ($K_D^{apt}$),

$$K_D^{apt} = \frac{[U][T]}{[B]} = \frac{k_{off}^{apt}}{k_{on}^{apt}}. \quad (2)$$

Target binding to the aptamer depletes the unfolded state, shifting the quenching reaction towards the unfolded state, generating more signal. Assuming a quenching efficiency of $\eta$, this equilibrium shift generates a target concentration-dependent signal given by

$$S_{eq} = [apt]_{total} \frac{1 + (1-\eta)K_Q + \frac{[T]}{K_D^{apt}}}{1 + K_Q + \frac{[T]}{K_D^{apt}}}. \quad (3)$$

The effective dissociation constant ($K_D^{eff}$), which reflects the effective binding affinity of the overall equilibrium, can be

derived[16] as

$$K_{\mathrm{D}}^{\mathrm{eff}} = K_{\mathrm{D}}^{\mathrm{apt}}(1 + K_{\mathrm{Q}}). \qquad (4)$$

Thus, while $K_{\mathrm{D}}^{\mathrm{eff}}$ is bounded by the binding properties of the native aptamer ($K_{\mathrm{D}}^{\mathrm{apt}}$), it also depends strongly on $K_{\mathrm{Q}}$. Moreover, the background signal of the construct is strongly related to $K_{\mathrm{Q}}$ by

$$S_{\mathrm{background}} = \frac{[\mathrm{apt}]_{\mathrm{total}}}{1 + K_{\mathrm{Q}}}. \qquad (5)$$

Therefore, it is crucial to understand the relative contributions of $L_{\mathrm{DS}}$ and $L_{\mathrm{loop}}$ to $K_{\mathrm{Q}}$. We isolate the effects of these independent tuning mechanisms by considering $K_{\mathrm{Q}}$ to be given by

$$K_{\mathrm{Q}} = \frac{[DS]_{\mathrm{eff}}}{K_{\mathrm{D}}^{\mathrm{DS}}}, \qquad (6)$$

where $[DS]_{\mathrm{eff}}$ is the effective concentration of the displacement strand that arises from covalent coupling to the native aptamer—a function of $L_{\mathrm{loop}}$—and $K_{\mathrm{D}}^{\mathrm{DS}}$ represents the dissociation constant for the hybridization of an unlinked displacement strand—a function of $L_{\mathrm{DS}}$. $[DS]_{\mathrm{eff}}$ constitutes the entropic component of $K_{\mathrm{Q}}$, whereas $K_{\mathrm{D}}^{\mathrm{DS}}$ constitutes the enthalpic component of $K_{\mathrm{Q}}$. If the displacement strand is too short (high $K_{\mathrm{D}}^{\mathrm{DS}}$) or the linker is too long (low $[DS]_{\mathrm{eff}}$), $K_{\mathrm{Q}}$ will be small, resulting in a large background (Eq. (5)) and little signal change upon the addition of target (Eq. (3)).

We assume that $K_{\mathrm{D}}^{\mathrm{DS}}$ is related to the binding energy of the free, untethered displacement strand, $\Delta G_{\mathrm{DS}}$:

$$K_{\mathrm{D}}^{\mathrm{DS}} = \exp\left(\frac{\Delta G_{\mathrm{DS}}}{RT}\right), \qquad (7)$$

where $\Delta G_{\mathrm{DS}} \cong L_{\mathrm{DS}} * -1.7\,\mathrm{kcal\,mol^{-1}\,bp^{-1}}$[25]. We can also make arguments for the approximate scaling of $K_{\mathrm{Q}}$ with $L_{\mathrm{loop}}$ based on observations of rates of DNA hairpin closure as a function of loop size. Since the dissociation rate ($k_{\mathrm{off}}^{\mathrm{DS}}$) is relatively independent of $L_{\mathrm{loop}}$ and the association rate ($k_{\mathrm{on}}^{\mathrm{DS}}$) has been shown to scale inversely with $L_{\mathrm{loop}}$ to the power of $2.6 \pm 0.3$[26], we can approximate that $K_{\mathrm{Q}} = \frac{k_{\mathrm{on}}^{\mathrm{DS}}}{k_{\mathrm{off}}^{\mathrm{DS}}}$ scales as $\sim L_{\mathrm{loop}}^{-2.6}$. Therefore, increases in $L_{\mathrm{DS}}$ or decreases in $L_{\mathrm{loop}}$ will be mirrored by an increase in $K_{\mathrm{Q}}$, shifting the equilibrium towards the quenched state, which results in decreased background signal at the expense of higher $K_{\mathrm{D}}^{\mathrm{eff}}$. Moreover, we can derive that $L_{\mathrm{loop}}$ will have a subtler per-base effect on $K_{\mathrm{Q}}$ relative to $L_{\mathrm{DS}}$ and that increases in $L_{\mathrm{loop}}$ will have diminishing returns because $\frac{dK_{\mathrm{Q}}}{dL_{\mathrm{loop}}}$ decreases as $\frac{1}{L_{\mathrm{loop}}}$ (Supplementary Note 1).

This qualitative reasoning also yields testable insights into the kinetic control of the system. We first need to discuss how changes in the individual rates manifest as changes in the observed rate. The complete binding kinetics of the population shift mechanism[27] are derived in Supplementary Note 2. The observed kinetics of the ISD construct will always be an additive mixture of two exponential responses with fast and slow kinetic rates. An in-depth discussion of the limits that the kinetics of the native aptamer impose on the kinetics of the ISD switching response is provided in Supplementary Note 3. Decreasing $L_{\mathrm{loop}}$ increases $k_{\mathrm{on}}^{\mathrm{DS}}$ with little effect on $k_{\mathrm{off}}^{\mathrm{DS}}$ and decreasing hybridization strength (i.e., decreasing $L_{\mathrm{DS}}$) increases $k_{\mathrm{off}}^{\mathrm{DS}}$ much more than $k_{\mathrm{on}}^{\mathrm{DS}}$[26]. Since the observed binding rates ($k_{\mathrm{obs}}^{\mathrm{fast}}$ and $k_{\mathrm{obs}}^{\mathrm{slow}}$) depend on the sum of $k_{\mathrm{on}}^{\mathrm{DS}}$ and $k_{\mathrm{off}}^{\mathrm{DS}}$ (Supplementary Equations (12) and (13)), we can determine that decreasing $L_{\mathrm{loop}}$ or $L_{\mathrm{DS}}$ will both increase $k_{\mathrm{obs}}$. To summarize our qualitative findings, we expect that $L_{\mathrm{DS}}$ and $L_{\mathrm{loop}}$ will have opposing effects on ISD switch thermodynamics but additive effects on the kinetics, and that $L_{\mathrm{DS}}$ will

have a more profound impact per base than $L_{\mathrm{loop}}$ on effective binding affinity.

**Experimental characterization of the ISD switch.** In order to experimentally validate the predictions of this model, we characterized ligand binding for an array of ISD switches (Supplementary Fig. 2) derived from the same ATP aptamer[20]. We introduced displacement strands with $L_{\mathrm{DS}}$ ranging from 5 to 9 base pairs (bp) and poly-T linkers of various lengths to yield $L_{\mathrm{loop}}$ ranging from 23 to 43 nucleotides (nt). Using a plate reader-based assay, we observed the fluorescence change as a function of ATP concentration. As expected, increasing $L_{\mathrm{DS}}$ with a constant $L_{\mathrm{loop}}$ resulted in decreased background signal and lower apparent affinity ($\uparrow K_{\mathrm{D}}^{\mathrm{eff}}$) (Fig. 3a). Fits of Supplementary Equation (1) (Supplementary Fig. 2) reveal a clear trend in which $K_{\mathrm{D}}^{\mathrm{eff}}$ increases with $L_{\mathrm{DS}}$, reflecting a reduction in effective binding affinity (Fig. 3b). We observed that $K_{\mathrm{D}}^{\mathrm{eff}}$ can be decreased up to 1,200-fold by removing three bases from the displacement strand (e.g., converting 9_23 to 6_23), with an average fold change of ~6.7 ± 2.4 per bp. Notably, the addition or removal of a single base from the displacement strand can shift $K_{\mathrm{D}}^{\mathrm{eff}}$ by more than an order of magnitude.

In contrast, we observed that changing $L_{\mathrm{loop}}$ has a subtler per-base effect on $K_{\mathrm{D}}^{\mathrm{eff}}$, with just a ~0.83 ± 0.15 fold change in $K_{\mathrm{D}}^{\mathrm{eff}}$ per additional base (Fig. 3c, d). On average, the linker must be increased by 17.7 ± 11.9 nt in order to shift the binding curve by an amount equivalent to the removal of a single base from the displacement strand. This loop/base equivalence value varies from construct to construct (Supplementary Table 2), but it is epitomized by the observation that adding 20 nt to the linker of 8_23 (generating 8_43) results in the same $K_{\mathrm{D}}^{\mathrm{eff}}$ as removing 1 bp from the displacement strand of 8_23 (generating 7_23) (Fig. 3e). The vast difference in these effects enables us to modulate effective binding affinity both finely (by tuning $L_{\mathrm{loop}}$) and over a wide functional range (by tuning $L_{\mathrm{DS}}$). Furthermore, in order to benchmark our ISD constructs with previous literature reporting duplexed aptamers (DAs)[12,22,24], we have also measured and characterized the corresponding constructs, in which an unlinked displacement strand is added to the solution at different concentrations (Supplementary Fig. 6). This change in concentration of unlinked displacement strand is a proxy for the change in effective concentration with different $L_{\mathrm{loop}}$.

We note that increasing $L_{\mathrm{DS}}$ changes the binding mechanism from independent to cooperative binding events. In contrast, changing loop length does not appear to change the binding mechanism. With short DS's, we observe a bimodal binding curve, whereas long displacement strands result in a single binding curve with cooperativity. Interestingly, the larger of the two $K_{\mathrm{D}}$'s remains constant while changing DS changes the smaller $K_{\mathrm{D}}$. This likely indicates that for long DS, cooperative binding of two ATPs is required to result in strand displacement. For short DS, it is possible that a single ATP can partially disrupt the DS on its own. It has also been suggested that for certain unlinked displacement strands, ATP binding can induce stand displacement in the ATP aptamer[24].

Next, we measured the temporal response of molecular recognition for all of our constructs to validate the previously described kinetic contributions of $L_{\mathrm{loop}}$ and $L_{\mathrm{DS}}$. Here, we injected ATP at known concentrations and observed the kinetics of the fluorescent response. As anticipated, we found that decreasing $L_{\mathrm{DS}}$ with a constant $L_{\mathrm{loop}}$ (Fig. 4a, b) or decreasing $L_{\mathrm{loop}}$ with a constant $L_{\mathrm{DS}}$ (Fig. 4c, d) resulted in faster temporal responses. By combining these two tuning mechanisms, we could vary the switching time constant ($k_{\mathrm{obs}}^{-1}$) by over 20-fold, ranging from ~3 s to ~170 ms. We note that even our slowest constructs represent a

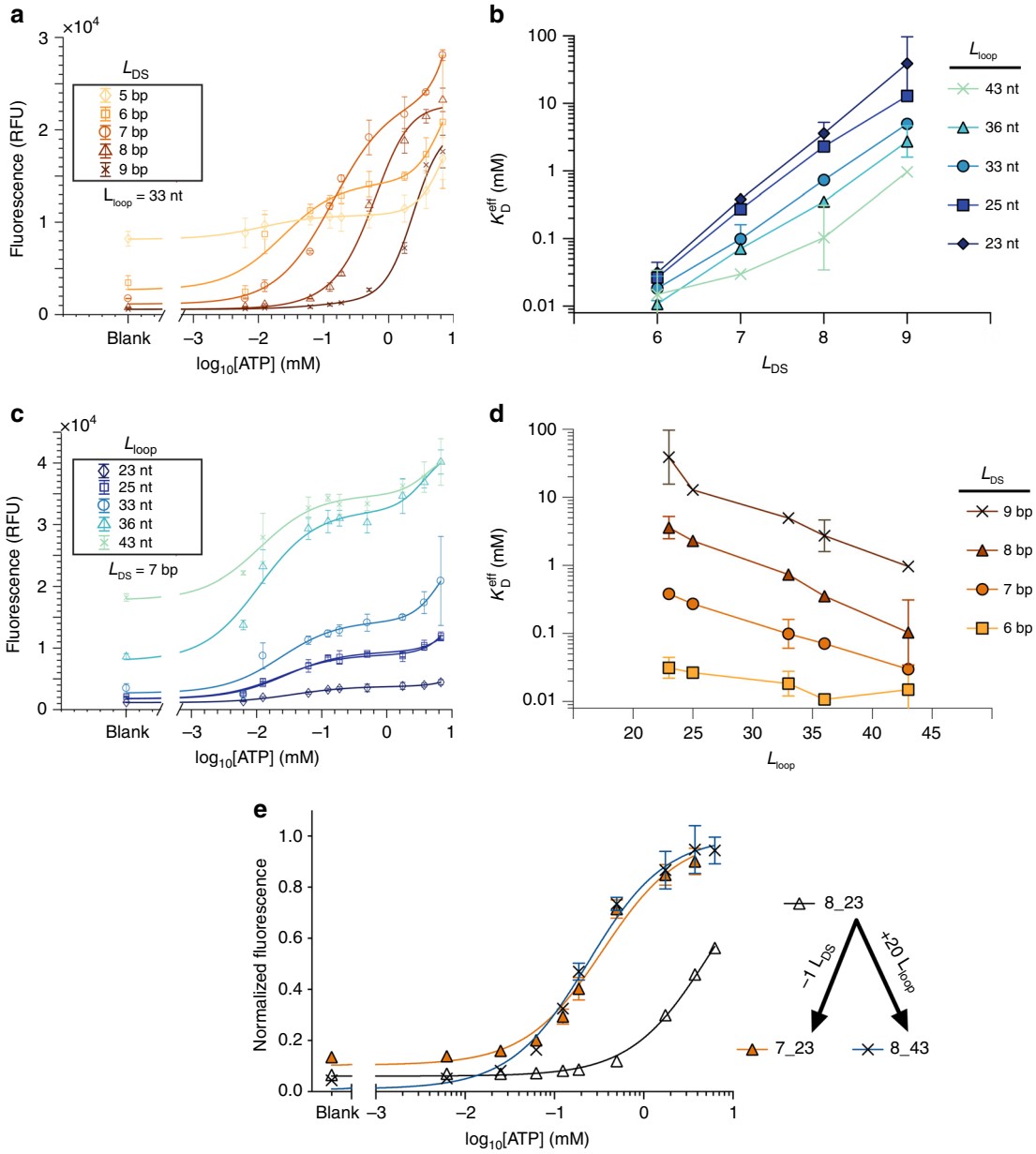

**Fig. 3** Binding curve modulation via ISD switch design. **a** Changing $L_{DS}$ from 5 to 9 bp while maintaining $L_{loop}$ at 33 nt shifts the binding curve to the right and reduces background signal. **b** $K_D^{eff}$ increases with $L_{DS}$ given a fixed $L_{loop}$. **c** Changing $L_{loop}$ from 23 to 43 nt while holding $L_{DS}$ constant at 7 bp shifts the binding curve to the left and increases background signal. **d** $K_D^{eff}$ decreases with $L_{loop}$ given a fixed $L_{DS}$. **e** The removal of a single base from the displacement strand of 8_23, generating 7_23, causes the same binding curve shift as adding 20 bases to the linker of 8_23, generating 8_43. All plots are averaged over three replicates ($n = 3$). All trials used [aptamer] = 250 nM. Error bars in **a**, **c**, and **e** represent the standard deviation of the average; error bars in **b** and **d** represent the error calculated via propagation of errors (Methods). Fits in **a** and **c** were conducted according to Supplementary Equation (1) (Supplementary Fig. 2). **a** and **c** show raw fluorescence data, whereas panel **e** is normalized by a single-site hyperbolic binding curve. Fit parameters for constructs in which $L_{DS} = 5$ have been omitted from **b** and **d** because we were unable to obtain robust fits for $K_D^{eff}$. Raw thermodynamic plots for all ISD constructs are provided in Supplementary Fig. 5. Source data are provided as a Source Data file

marked improvement over traditional aptamer beacons, which typically exhibit time constants on the order of minutes to hours[18,28]. The fast kinetics of the ISD switch are attributable to a much higher $k_{on}^{DS}$ resulting from the high effective concentration of the displacement strand when tethered to the aptamer. This high $k_{on}^{DS}$ allows us to use much shorter displacement strands than are possible with aptamer beacons, which in turn results in a much faster $k_{off}^{DS}$. The simultaneous increase in both $k_{on}^{DS}$ and $k_{off}^{DS}$ greatly increases the observed rate. Indeed, switches with $L_{DS} = 5$ bp achieved temporal responses exceeding the time resolution of

our detector; with a time delay of 465 ms between injection and measurement, $k_{obs}^{MAX}$ is ~10 s$^{-1}$.

**Decoupling thermodynamics and kinetics.** Our thermodynamic and kinetic findings suggest the possibility of designing ISD switches such that temporal resolution can be tuned completely independently of binding affinity. Since the effective affinity of our construct depends on the equilibrium constant for hairpin formation, $K_Q = \frac{k_{on}^{DS}}{k_{off}^{DS}}$, it is clear that the binding kinetics can be increased while maintaining the same effective binding affinity

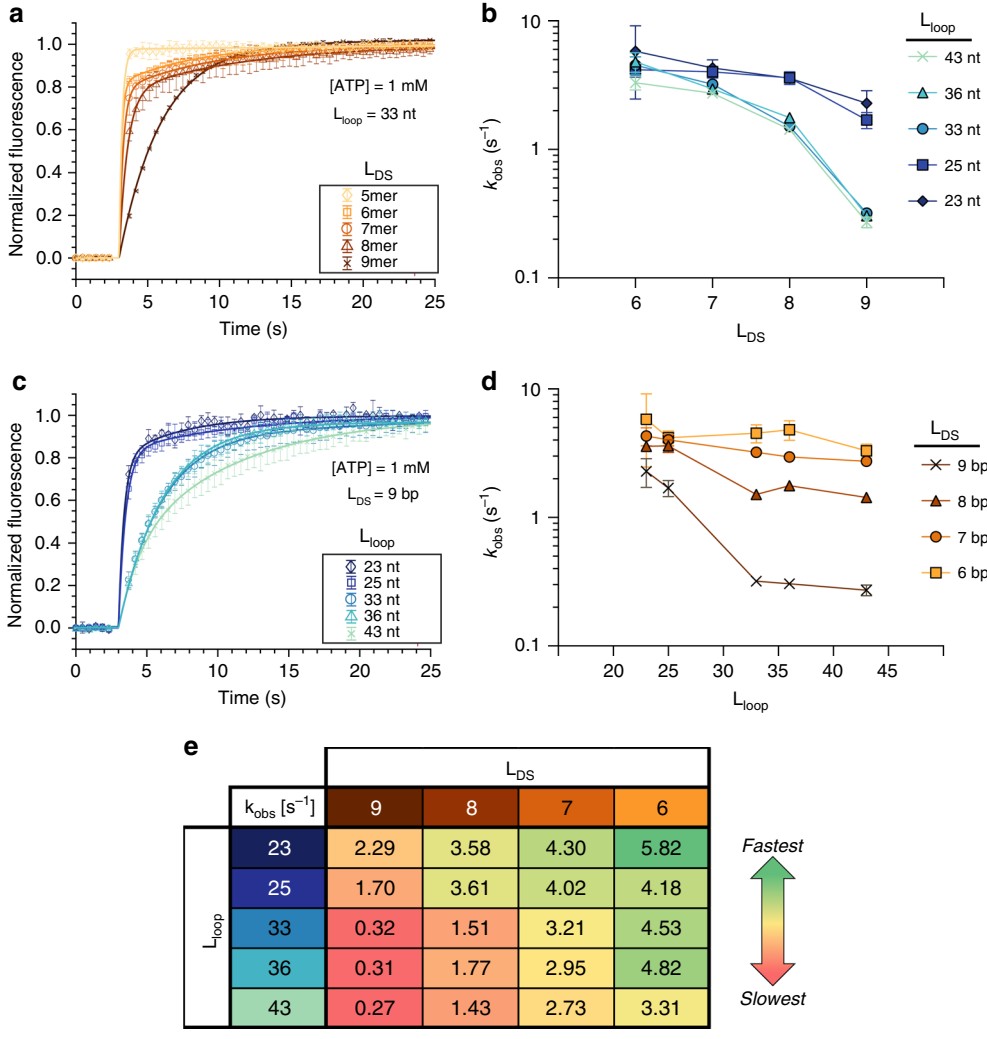

**Fig. 4** Temporal response modulation via ISD switch design. **a** Normalized signal change upon injection of 1 mM ATP. Increasing $L_{DS}$ while holding $L_{loop}$ constant at 33 nt results in slower kinetics. **b** Effect on $k_{obs}$ of increasing $L_{loop}$ in an ISD construct with constant $L_{DS}$ in the presence of 1 mM ATP. **c** Normalized signal change upon injection with 1 mM ATP. Increasing $L_{loop}$ while keeping $L_{DS}$ constant at 9 results in slower kinetics. **d** Effect on $k_{obs}$ of increasing $L_{DS}$ in an ISD construct with constant $L_{loop}$ in the presence of 1 mM ATP. **e** $k_{obs}$ as a function of both $L_{DS}$ and $L_{loop}$ after injection with 1 mM ATP. Sequences with $L_{DS} = 5$ had switching responses faster than the time resolution of our detector and could not be fit accurately and have thus been omitted from panels **b**, **d**, and **e**. Error bars in **a** and **c** represent standard deviation ($n = 3$), whereas those in **b** and **d** represent the 95% confidence intervals in the fit parameter. Confidence intervals for **e** are listed in Supplementary Fig. 3. Source data are provided as a Source Data file

provided the ratio between the association and dissociation rates of the displacement strand is preserved. Decreasing $L_{loop}$ increases the binding rate and decreases the effective affinity of our construct. This decrease in affinity can be offset by shortening $L_{DS}$, which in turn results in a net additive increase in temporal resolution. Based on the observed dependencies of our two control parameters, we hypothesized that it should be feasible to achieve faster switching responses and maintain effective affinity by decreasing $L_{loop}$ and $L_{DS}$ simultaneously.

We confirmed this prediction with three pairs of constructs (Fig. 5a) that each have statistically indistinguishable effective affinities (Fig. 5b). Constructs with both a longer loop length and a stronger hairpin had universally slower temporal resolution (Fig. 5c). Tandem tuning of the two parameters allowed us to increase the temporal response by up to 6-fold without changing $K_D^{eff}$. For example, in pair I, decreasing $L_{loop}$ from 36 to 23-nt increases $k_{on}^{DS}$, while decreasing $L_{DS}$ from 9 to 8-bp increases $k_{off}^{DS}$, resulting in much more rapid observed kinetics with a roughly constant $K_Q$. Importantly, we were able to achieve this tunability over a wide range of effective affinities (~90 μM to ~5.8 mM).

**Precision tuning through DS complementarity**. We hypothesized that even finer control over the hybridization strength of the duplexed region of the ISD construct would be possible if we manipulate the complementarity of the two strands. Previous work has shown that incorporating mismatches into a displacement strand increases the sensitivity of the resulting biosensor[29]. Indeed, by expanding our design space to include single-base mismatches, we calculated that we could increase the theoretical resolution of our tuning capability by >10-fold relative to that of perfectly-matched displacement strands (Supplementary Fig. 7a). Since the introduction of mismatches has been shown to drastically increase $k_{off}$ and $k_{on}$ for two hybridizing strands[30], we anticipated that the introduction of mismatches would greatly increase the observed kinetics. Thus, mismatches should enable finer enthalpic control over the binding curve and enhance our ability to increase kinetics independently of affinity.

To experimentally confirm these predictions, we introduced single mismatches of different identities (A, G, C or T) at various positions throughout the displacement strands of three constructs: 8_33 ($L_{DS} = 8$, $L_{loop} = 33$), 9_33 ($L_{DS} = 9$, $L_{loop} = 33$),

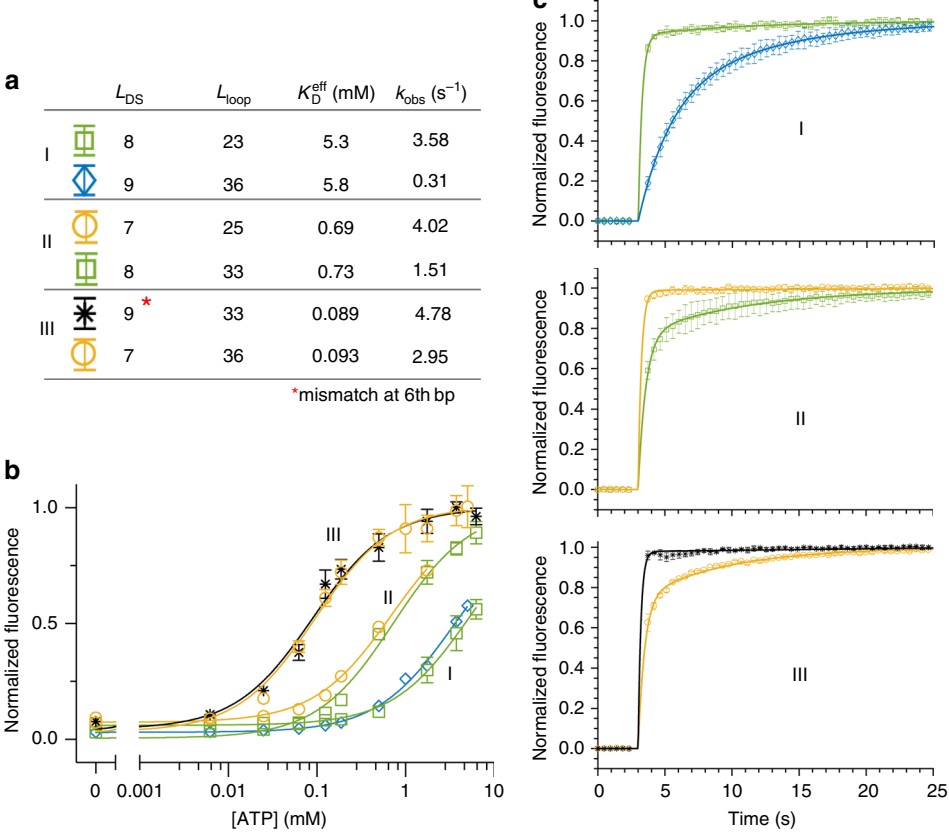

**Fig. 5** Independent tuning of kinetics and thermodynamics. Simultaneously increasing both $L_{DS}$ and $L_{loop}$ has synergistic effects on temporal resolution but opposing effects on $K_D^{eff}$. Therefore, it should be possible to change the kinetic response of the switches while holding the affinity constant. **a** We confirmed this by examining three pairs (I, II, III) of ISD constructs. **b** Each pair exhibited nearly identical binding curves, **c** but has been engineered in terms of $L_{DS}$ and $L_{loop}$ to exhibit vastly different kinetic responses. The $K_D^{eff}$ listed in **a** and the binding curve fits in **b** are respectively calculated from and normalized to a single-site hyperbolic binding curve. For kinetics measurements, pair I was run at [ATP] = 2.5 mM and pairs II and III were run at [ATP] = 1 mM. Plots were averaged over three replicates ($n = 3$) and error bars represent the standard deviation. Source data are provided as a Source Data file

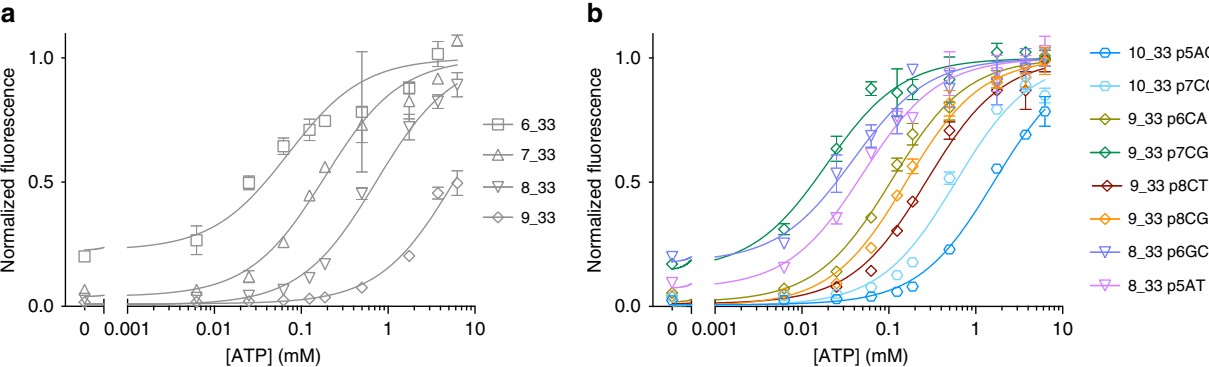

**Fig. 6** Fine tuning of binding curves via the incorporation of mismatches in the displacement strand. **a** Modulating affinity via $L_{DS}$ alone leads to huge jumps in affinity. **b** In contrast, the incorporation of single mismatches into the displacement strand produces much more granular shifts in effective binding affinity. Mismatches are described such that 9_33 p6CG indicates that position 6, as defined from the 3'- end, was changed from C to G. Curves were averaged ($n = 3$) and were fit to and normalized by a single-site hyperbolic binding curve. Source data are provided as a Source Data file

and 10_33 ($L_{DS} = 10$, $L_{loop} = 33$). Upon comparing the thermodynamic properties of the original constructs to those containing the mismatches, we found that we were able to obtain more closely spaced binding curves by altering the position and identity of the mismatch. For constructs with a 33-nt loop, for example, the average fold change between the set of $K_D^{eff}$ values that are obtained from perfectly matched displacement strands was $6.55 \pm 1.01$ (Fig. 6a). However, using just a small subset of

possible mismatches, we were able to reduce this average spacing to $1.72 \pm 0.34$ (Fig. 6b; Supplementary Fig. 7b). Therefore, by modulating the position and identity of mismatches, we can generate sets of constructs with much finer enthalpic control than would be possible by changing $L_{DS}$ alone. Lastly, the incorporation of mismatches substantially increases $k_{off}^{DS}$[30], such that mismatches not only confer greater control over the thermodynamics but also dramatically increase temporal resolution

relative to perfectly-matched displacement strands (Supplementary Fig. 7c).

## Discussion

Although aptamer-based molecular switches are powerful tools in biotechnology, their utility has been constrained by a limited ability to rationally engineer their binding characteristics in terms of affinity and kinetics. Prior studies have indicated that the thermodynamics and kinetics of such switches are coupled in such a manner that gains in one parameter will result in sacrifices in the other[18]. Here, we have demonstrated an aptamer switch design that allows remarkably precise independent control of its thermodynamic and kinetic parameters. Our ISD construct connects an existing aptamer to a partially complementary displacement strand via a poly-T linker, such that alterations in the length of either feature can meaningfully shift the equilibrium binding to the aptamer's target. We used a mathematical model to demonstrate how changes in the hybridization strength of the displacement strand would confer coarse control over switch affinity; at the same time, changes in linker length produce a subtler effect per added or removed base. We subsequently confirmed this expectation experimentally and demonstrated the capacity to carefully manipulate the binding characteristics of our switch through these two parameters. For example, we can increase binding kinetics by an order of magnitude with minimal effect on aptamer affinity by selectively shortening both $L_{DS}$ and $L_{loop}$. Furthermore, we have shown that even finer tuning of ISD binding properties is possible when we manipulate the strength of displacement strand hybridization through the targeted introduction of individual base-mismatches into the displacement strand sequence.

As the desire for rapid molecular detection becomes more prevalent, so too will the need to tune the kinetics of molecular recognition independently of binding affinity. Our approach is advantageous in this regard, as it offers opportunities for control that exceed those of existing molecular switch designs, which are generally constrained by tight coupling of kinetic and thermodynamic parameters and offer less freedom for structural manipulation. We have demonstrated the feasibility of achieving ultrafast kinetic responses (on the order of hundreds of milliseconds) with our ISD constructs without meaningfully sacrificing target affinity, whereas aptamer beacons typically exhibit kinetics on the order of minutes to hours. Critically, our molecular switch design should be compatible with virtually any aptamer sequence, making it feasible to design optimized molecular switches that are ideally suited for a diverse array of biotechnology and synthetic biology applications.

## Methods

**Reagents**. All chemicals were purchased from Thermo Fisher Scientific unless otherwise noted, including ATP (25 µmol, 100 mM), Tris-HCl Buffer (1 M, pH 7.5), magnesium chloride (1 M, 0.2 µm filtered), and Hyclone molecular biology-grade water (nuclease-free). Oligonucleotides modified with Cy3 fluorophore at the 5′ ends and DABCYL quencher at the 3′ ends, purified by HPLC, were purchased from Integrated DNA Technologies. All sequences used in this work are shown in Supplementary Table 1. All oligonucleotides were resuspended in nuclease-free water and stored at −20 °C. All experiments were performed in triplicate experiments unless otherwise noted.

**Measurements of effective binding affinity**. To obtain binding curves, 40 µL reactions were prepared in 1× ATP binding buffer (10 mM Tris–HCl, pH 7.5 and 6 mM MgCl$_2$) with 250 nM aptamer and final ATP concentrations in the range of 6.25 µM to 6.75 mM. The fluorescence spectra for all samples were measured at 25 °C on a Synergy H1 microplate reader (BioTeK). Emission spectra were monitored in the 550–700 nm range with Cy3 excitation at 530 nm and a gain of 100, in 96-Well Half Area Plates (Corning™ 96-Well Half Area Black Flat Bottom Polystyrene Microplates). All measurements were performed in triplicate. Representative concentration-dependent emission spectra are shown in Supplementary Fig. 1.

**Measurements of binding kinetics**. ISD constructs of varying linker and displacement strand lengths were suspended at a concentration of 333.3 nM in a 30 µL total volume of 1x ATP binding buffer (10 mM Tris-HCl, pH 7.5 and 6 mM MgCl$_2$). Kinetic fluorescence measurements of the quencher-fluorophore pair were made using a Synergy H1 microplate reader. Cy3 was excited at 530 nm, and unquenched fluorescence was measured at 570 nm emission using monochromators at the minimum possible regular time interval of 0.465 s. After timed injection of 10 µL ATP (final [ATP] = 0, 1, or 2.5 mM) in 1x ATP binding buffer into the 30 µL ISD solution, we measured the kinetic response in the 40 µL sample volume. We first normalized all kinetic data relative to the 0 µM target concentration to account for the effect of sample volume change upon injection with ATP. For plotting, we normalized the curves to range from 0 to 1 in order to visually emphasize changes in rate constants rather than the background and peak levels that are dictated by the thermodynamics. All measurements were performed in triplicate.

**Thermodynamic analysis**. Three replicates of inputs ($X = \log([ATP])$ in mM) versus outputs ($Y$ in raw RFU intensity) were fit individually for each construct to extract the effective binding affinity. The resultant parameters from fitting $X$ and $Y$ to Supplementary Equation (2) were averaged over the three independent fits. We fit the logarithmic values of thermodynamic constants $pK_{D1} = -\log(K_{D1})$, $pK_{D2} = -\log(K_{D2})$, and $pK_Q = -\log(K_Q)$ such that Supplementary Equation (2) becomes:

$$Y = B_{max} \frac{10^{-pK_{D1}}10^{-pK_{D2}} + \eta_1 10^{-pK_{D1}}10^X + \eta_2 10^{-pK_{D2}}10^X + 10^{-pK_{D1}}10^{-pK_{D2}}10^{2X}}{10^{-pK_{D1}}10^{-pK_{D2}}(1 + K_Q) + 10^{-pK_{D1}}10^X + 10^{-pK_{D2}}10^X + 10^{-pK_{D1}}10^{-pK_{D2}}10^{2X}}$$

(9)

Fits were performed via MATLAB's *lsqcurve* function with initial guesses equal to

$B_{max}^{guess} = \max(Y) - \min(Y)$, $\eta_1^{guess} = 1$, $pK_{D1}^{guess} = -X\left(Y \sim \frac{\max(Y)+\min(Y)}{2}\right)$, $\eta_2^{guess} = 1$, $pK_{D2}^{guess} = pK_{D1}^{guess} - 6$, and $pK_Q^{guess} = 1 - \frac{\max(Y)-\min(Y)}{2\min(Y)}$.

Upper bounds were set to $B_{max}^{upper} = \max(Y) * 10$, $\eta_1^{upper} = 1.01$, $pK_{D1}^{upper} = 1$, $\eta_2^{upper} = 1.01$, $pK_{D2}^{upper} = 10$, and $pK_Q^{upper} = 10$.

Lower bounds were set to $B_{max}^{lower} = \frac{\max(Y)}{2}$, $\eta_1^{lower} = 0$, $pK_{D1}^{lower} = -6$, $\eta_2^{lower} = 0$, $pK_{D2}^{lower} = -3$, and $pK_Q^{lower} = -10$.

Fitting was performed using a maximum of 100,000 iterations. $pK_{D1}$ and $pK_Q$ were averaged over at least three replicate fits for each construct. The effective binding affinity, $K_D^{eff}$, was then calculated by

$$K_D^{eff} = 10^{-pK_{D1}}\left(1 + 10^{-pK_Q}\right),$$

(10)

with the standard error given by propagation of errors

$$\sigma_{K_D^{eff}} = \sqrt{\left(\ln(10)\sigma_{pK_{D1}}10^{-pK_{D1}}(1 + 10^{-pK_Q})\right)^2 + \left(\ln(10)\sigma_{pK_Q}10^{-pK_{D1}}10^{-pK_Q}\right)^2}.$$

(11)

The curves plotted in Fig. 3a and c were fit to the average of the three replicates.

**Normalized thermodynamic plots**. For ease of comparison, the data in Figs. 3e, 5b, and 6 were fit to and normalized according to a single-site hyperbolic binding curve:

$$y = (B_{max} - y_0)\frac{x}{x + K_D^{eff}} + y_0.$$

(12)

For Figs. 3e, 5b, and 6, the average fluorescence value was divided by $B_{max}$ prior to plotting. Error bars were reported via propagation of errors for the standard deviation of the average fluorescence and the error in the fit for $B_{max}$.

**Kinetic analysis**. Kinetic data were first normalized to a zero ATP control to account for changes in volume due to the injection of ATP. We fit each replicate individually to

$$y(t) = \begin{cases} A - B\exp(-k_1 t) - C\exp(-k_2 t) & t > t_0 \\ A - B - C & t \leq t_0 \end{cases},$$

(13)

where $k_1 > k_2$.

Each replicate was then normalized to a range of zero to one by:

$$y^*(t) = \frac{y - A + B + C}{N}$$

(14)

where

$$N = \begin{cases} B & k_2 \leq 0.015\, s^{-1} \\ B + C & k_2 > 0.015\, s^{-1} \end{cases}.$$

(15)

A piecewise function was used for N to control for an artifact of the fitting function, wherein if there is no observable $k_2$, the fit function still forces the fit to $k_2$ which can result in extremely large values of C. Therefore, we omit C from the normalization if $k_2$ is very slow. The normalized responses were averaged, and the average response was again fit to Eq. (12) to obtain a rate representative of all three

trials. Rate constants are reported as the best fit values ± 95% upper/lower confidence intervals.

## Data availability

All data underlying the findings of this study are available from the authors upon reasonable request. The source data underlying Figs. 3a, c, e, 4a, c, 5b, c, 6a, b, Supplementary Figs. 1, 2, 5, 6, and 7, and Supplementary Table 2 are provided as a Source Data file.

## Code availability

The *mathematica* code (version 11.1.1.0) containing the kinetic derivation is provided at https://github.com/btotherad77/isd.

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

## Acknowledgements

This work was supported by NIH SPARC Initiative (OT2OD025342), NIH (1R01GM129313), Wu Tsai Neurosciences Institute at Stanford, Stanford Maternal and Child Health Research Institute and the Chan-Zuckerberg Biohub. H.T.S. is a Chan Zuckerberg Biohub investigator. We are thankful to Philips Healthcare and NSERC PDF (A.A.H.) and the Medtronic Foundation Stanford Graduate Fellowship (I.A.P.T). We also thank Dr. Evelin Sullivan of the Technical Communications Program at Stanford for her thoughtful comments and edits on the manuscript.

## Author contributions

B.D.W., A.A.H. and H.T.S. conceived the initial concept. B.D.W., A.A.H., and I.A.P.T. designed the experiments. A.A.H., I.A.P.T. and B.D.W. executed the experiments. B.D.W. developed the model and analyzed the data. B.D.W., M.E. and H.T.S. wrote the manuscript. All authors edited, discussed, and approved the whole paper.

## Competing interests

The authors declare no competing interests.
