## [Peer Review File · Nature Communications]

Reviewers' comments:

Reviewer #1 (Remarks to the Author):

Summary

This work explored the thermodynamic and kinetic behavior of a common class of aptamer switches designed using an intramolecular strand-displacement (ISD) strategy. The aptamer switch studied here consisted of a DNA aptamer for the target ATP, which was appended with a flexible linker (of variable length) that tethers a short oligonucleotide (the displacement strand, DS, also of variable length). The DS is designed to hybridize to a section of the aptamer sequence, and this construct generates ATP-dependent fluorescence.

The authors sought to understand how the lengths of the linker and displacement strand impact the thermodynamic and kinetic behavior of the aptamer switch, with the goal of understanding if the design properties could be used to independently tune aptamer performance. By changing the length of the DS, and the length of the linker between the aptamer and the DS (thereby changing the effective concentration of the DS), together with the introduction of mismatched bases in the DS, the authors demonstrated that the thermodynamic and kinetic properties of aptamer switch can be independently and rationally tuned over a wide range.

Recommendation

In general, the work should be of high interest to the biosensor community. In particular, previous efforts on tuning aptamer biosensors required a compromise between reaction rate and affinity. This is because the thermodynamics and kinetics of ligand binding, structure-switching and subsequent signal generation of aptamer constructs are tightly coupled. This work introduced a generalizable design approach that allows for independent control of thermodynamic and kinetic properties of aptamer switches, which would enable more efficient development of biosensors.

The experiments performed are pertinent and the data presented support the authors' qualitative reasoning and anticipation of the behavior of the constructs. Publication in Nature Communications could be considered after addressing the a number of issues detailed in the comments below, which notably include conducting a set of experiments with trans-DA constructs to help address ambiguity in the interpretation, and help establish a better comparison to prior studies on this topic.

1. While the control of the thermodynamics is easy to understand and builds on past findings, the control over the kinetic properties is particularly interesting, however the reasoning of how the kinetics are controlled is less intuitive.

Specifically, Figure 4c shows that when LDS is held constant, decreasing Lloop leads to increasing kobs. Whereas a decrease in Lloop would be expected to increase konDS (higher effective concentration of the DS), the authors also claim that changes in Lloop result in negligible changes to koffDS, implying that an increase in konDS directly leads to an increase in kobs. However, it is not clear to this reviewer how an increase in konDS would directly increase kobs - rather, we find the claim of a faster konDS leading to faster kobs counter-intuitive. Given the experimental protocol followed in this work, aptamer switches with increased konDS (but equal koffDS) should respond more slowly to the introduction of ATP, since the DSs in these aptamers would more strongly outcompete ATP for any existing unfolded constructs, and by this reasoning the ISDs would require more time to reach an equilibrium state.

An alternative interpretation of the data might suggest that changes in Lloop are actually impacting koffDS, thereby leading to faster unbinding of the DS and more rapid binding of ATP. As the signal is generated upon DS dissociation, kobs should instead be largely dependent on koffDS. Based on this reasoning, Figure 4c therefore suggests koffDS changes with Lloop. One possible explanation for this change in koffDS would be strain to the structure of the aptamer switch that is caused by the use of a short linker. Following this logic, shorter linkers would lead to higher strain

in the structure, thereby causing a faster release of the DS, and hence faster kinetics. A longer linker would confer higher flexibility or structural plasticity, requiring the DS to gradually “zip off” the aptamer, resulting in a slower reaction rate. Indeed, many factors are in play and determine the kinetics of the reaction. Hence, we recommend that the authors discuss the above logic and interpretation (or further clarify their claim that increases in k_{onDS} would directly increase k_{obs} , perhaps using realistic values in the derived equations provided, together with a logical interpretation).

2. The shape of the binding curves, especially for constructs with short LDS, show more complex binding behavior (e.g. Figure S5), such as the bimodal behavior described in equation S1. Given that the DSs used in this study occupy ATP binding site II in the aptamer, and that DSs hybridizing to this region of the aptamer have been shown to modulate ligand binding cooperativity (e.g. Armstrong & Strouse, *Bioconjugate Chem.*, 2014, 25, 1769-1776, and Munzar et al, *Chem. Sci.*, 2017, 8, 2251-2256), further discussion of the binding curves and of the possible ligand binding modes of these ISD constructs is merited.

3. To bolster the work presented here, we suggest that the authors compare the DSs with various Loop used in this study to the behavior of the same bi-molecular aptamer switches without a linker (i.e. trans-DAs, see Munzar et al., *Chem. Soc. Rev.*, 2019, 48, 1390-1419), which is a construct that has been more commonly used in the literature. A trans-DA contains a DS that is not tethered to the aptamer, and hence is effectively an ISD switch with an infinitely long and flexible linker (in fact, the concentration of free DSs could be used to mimic the effective linker distance, which would be interesting to demonstrate). With the addition of 8- and 9-mer trans-DAs, it would be expected that trans-DAs would have enhanced thermodynamics, with a decreased apparent K_D because the equilibrium is shifted towards DS-free species. In addition to expanding on the results presented here, these trans-DAs would also allow this work to be directly compared with more constructs (ATP-specific and otherwise) studied by others previously.

4. Some key references dealing with ATP aptamer probe designs should be included and discussed where appropriate, such as:

Armstrong and Strouse, *Bioconjugate Chem.*, 2014, 25, 1769-1776 (ISD design and binding dynamics)

Das et al, *Nat. Chem.*, 2012, 4, 642-648 (Use of mismatched DSs to improve aptamer probe performance)

Munzar et al, *Chem. Sci.*, 2017, 8, 2251-2256 (Induced-fit binding dynamics of trans-DAs)

Minor edits

- Figure 4 caption, Line 2: Increasing “Loop” while holding “LDS” constant should be Increasing “LDS” while holding “Loop”.
- Figure 1 : The text of the binding constants is garbled in the boxes.
- The method used for the measurement does not appear in the main text. It would clarify the reading to add a brief explanation of the protocol used.

Review by Dr. Jeffrey Munzar, Dr. Andy Ng and Dr. David Juncker

Reviewer #2 (Remarks to the Author):

The manuscript #214854 reports on a method for the fine-tuning of thermodynamics and kinetics of aptamer switches in an independent way, using a well known ATP DNA aptamer to prove the concept. It is a general strategy, which combines in a smart way previously explored solutions for controlling the enthalpic and entropic contributions of the recognition by such a type of synthetic receptors. It is based on the elongation of the aptamer sequence with a poly-T sequence with variable length, and a strand also of variable length and partially complementary to the aptamer, in such a way that the aptamer target induces an internal strand-displacement event. Authors demonstrate that the length of the poly-T linker and the length of the complementary strand can

be rationally designed to tune the affinity and kinetics of the target-aptamer binding, and develop a mathematic model to predict such an effect. The paper is well written and structured, the experiments are carefully designed and the results convincing. The strategy could find broad applications in biosensing, imaging or diagnostics. I recommend publishing the manuscript in Nature Communications with just minor considerations:

1. How the affinity of the parent aptamer influences the extent and magnitude of the window in which the effective affinity of the switches can be modulated.
2. A similar question arises from a kinetic point of view. Does the kinetics of the original aptamer have any influence on the range of values in which the observed kinetics can be controlled?
3. Is there any rational limit in the lengths of both the loop and the displacement strand to control the affinity and the temporal resolution of the designed aptamer switches?

Reviewer #1: We greatly appreciate the reviewers' efforts and their thorough consideration of our work. Although the review was generally enthusiastic and recognized the novelty and importance of the work, it raised a number of important questions and points of clarification that have been addressed in the revised manuscript. We believe that the manuscript is significantly stronger for it.

1. The reviewers note that our claim that increasing $k_{\text{on}}^{\text{DS}}$ will increase k_{obs} is counter-intuitive, arguing that the opposite could be true. The reviewers also provide an alternative explanation that decreasing loop length increases $k_{\text{off}}^{\text{DS}}$ instead of increasing $k_{\text{on}}^{\text{DS}}$. The reviewers suggest that we either clarify our argument or incorporate their rationale.

We appreciate the reviewers' careful consideration of our claims and their presentation of an alternative interpretation of the data. However, we respectfully disagree and endeavor here to adequately explain why. There are two main points raised by the reviewers:

- *How does changing loop length affect $k_{\text{on}}^{\text{DS}}$ and $k_{\text{off}}^{\text{DS}}$?*

In the paper, we state that decreasing loop length increases $k_{\text{on}}^{\text{DS}}$ because of an increased collision frequency due to the increased effective concentration. However, the reviewers posit that decreasing loop length would instead increase the off-rate of the displacement strand due to greater loop strain, which would better describe the observed data.

First, we apologize for not being completely clear in our explanation. The off-rate *will* change with loop length, but this change is negligible when compared to the change in the on-rate associated with a change in loop length. Prior work supports our explanation that decreasing loop length increases $k_{\text{on}}^{\text{DS}}$; for example, in Ref. 24 (<https://www.pnas.org/content/pnas/95/15/8602.full.pdf>), the authors examined the opening and closing rates of DNA hairpins with variable loop lengths (figure reproduced below), and found that the opening rate of the hairpin (*i.e.*, $k_{\text{off}}^{\text{DS}}$) was relatively insensitive to loop length (blue) compared to the closing rate (*i.e.*, $k_{\text{on}}^{\text{DS}}$), which changed as a strong function of loop length (green).

[redacted]

We believe this experimental evidence strongly favors our original rationale.

- Why would increasing k_{on}^{DS} increase k_{obs} ?

We agree that the statement that increasing k_{on}^{DS} increases k_{obs} is counter-intuitive. However, consider the simple case of the reaction $A + B \leftrightarrow AB$. If $[B] \gg [A]$, over time, the formation of $[AB]$ resolves to $[AB](t) = 1 - \exp\left(-\left(k_{on}[B] + k_{off}\right)t\right)$. Therefore $k_{obs} = k_{on}[B] + k_{off}$. This solution provides the seemingly paradoxical result that an increase in the off-rate results in an increase in the observed binding rate. The same outcome occurs in our scenario, albeit with a more complicated expression, in which the math produces two observed rate constants: k_{obs}^{fast} and k_{obs}^{slow} (**Supplemental Calculation 2**). Upon examining the solutions, we find that $\frac{dk_{obs}^{fast}}{dk_{off}^{DS}} \geq 0$ and $\frac{dk_{obs}^{slow}}{dk_{off}^{DS}} \geq 0$ for all positive values of k_{on}^{DS} , k_{on}^{apt} , k_{off}^{apt} , and $[T]$. Therefore, we conclude that increasing k_{off}^{DS} can only ever increase k_{obs} —or at least have no effect.

In addressing this comment, we also realized that a similar kinetic derivation has been previously published (<https://pubs.acs.org/doi/pdf/10.1021/bi3006913>), and have added the appropriate reference. Furthermore, we have added a thorough description of the limits of the kinetic tuning (**Supplementary Note 3**), as described below in the response to Reviewer #2's second comment.

2. The reviewers point out that many constructs – especially those with short displacement strands – exhibit more complex binding behavior than simple one-site binding. The reviewers requested further discussion of the binding curves and possible binding modes.

We appreciate the comment, but note that it is only applicable to two-site binding aptamers and is not necessarily applicable to the conceptual implementation of ISD in general. That being said, we agree that the observed phenomenon is interesting, and have included a discussion of the shape of the binding curves in the manuscript.

Based on our data, we don't know for sure what the exact binding mechanism is. However, we can speculate based on our observations. We note that increasing L_{DS} changes the binding mechanism from independent to cooperative binding events. On the other hand, changing loop length does not appear to change the binding mechanism. With a short displacement strand, we observe a bimodal binding curve, whereas a longer displacement strand results in a single binding curve with cooperativity. Interestingly, the larger of the two K_{DS} remains constant throughout, with changes in L_{DS} only changing the smaller K_D . This likely indicates that for a long displacement strand, cooperative binding of two ATPs is required for strand displacement. In contrast, for a shorter displacement strand, it is possible that a single ATP can partially disrupt displacement strand hybridization on its own. It has also been suggested that for certain unlinked displacement strands, ATP binding can induce strand displacement in the ATP aptamer. We have added these additional explanations to the revised manuscript.

3. The reviewers suggested including additional experiments to compare the ISD constructs to the corresponding unlinked, trans-DA constructs.

We thank the reviewers for this suggestion. We agree that these experiments allow readers to make more direct comparisons to previous literature, and that changes in the concentration of unlinked displacement strand will mimic changes in the effective concentration due to loop length. We have

confirmed this experimentally, and have added this information to the Supplemental Material. We would like to note that we avoided including this in the initial manuscript because our goal was to generate monolithic molecular switches to allow for long-lived integration with biosensors, which require reversibility under flow.

4. The reviewers suggested some additional references dealing with ATP aptamer probe designs.

We thank the reviewers for pointing out these articles, and have added these citations where appropriate:

- Armstrong and Strouse, *Bioconjugate Chem.*, 2014, **25**, 1769-1776
- Das *et al*, *Nat. Chem.*, 2012, **4**, 642-648
- Munzar *et al*, *Chem. Sci.*, 2017, **8**, 2251-2256

5. Figure 4 caption, Line 2: Increasing “Loop” while holding “LDS” constant should be Increasing “LDS” while holding “Loop”.

We have corrected this error in the text.

6. Figure 1: The text of the binding constants is garbled in Figure 1.

We have noted some compatibility issues when viewing figures on a PC, and have attached the figures as Powerpoint slides.

7. The method used for the measurement does not appear in the main text. It would clarify the reading to add a brief explanation of the protocol used.

We thank the reviewers for noting this, and have made the appropriate corrections.

Reviewer #2: We thank the reviewer for the enthusiastic review. Although the reviewer recommended the work for publication, s/he also raised a few opportunities for clarification.

1. The reviewer asked what limits the affinity of the parent aptamer places on the extent and magnitude of the window over which the effective affinity of the switch can be modulated.

This is an important question, and we apologize for not addressing this more clearly. We noted in the text that “while the effective affinity of our construct is defined by the binding properties of the native aptamer (K_D^{apt}), it also depends strongly on K_Q .” This is illustrated by equation (4): $K_D^{eff} = K_D^{apt}(1 + K_Q)$. We originally implicitly described this limit on the effective binding affinity, but we have modified the manuscript to state explicitly that K_D^{apt} places a bound on K_D^{eff} .

2. The reviewer also asked how the kinetics of the original aptamer limit the range of obtainable observed kinetics.

As with the thermodynamics, the kinetics of the native aptamer impose limits on the kinetic response of our ISD constructs. The observed kinetics of the ISD construct will always be an additive mixture of two exponential responses with fast and slow kinetic rates. Starting from our analytically determined rate constants for the induced fit model (**Supplementary Note 2**), we consider the two limiting cases where the kinetics of displacement strand binding have been tuned to be arbitrarily slower or faster than the kinetics of the native aptamer, such that $k_{off}^{DS}, k_{on}^{DS} \ll k_{off}^{apt}, [T]k_{on}^{apt}$ or $k_{off}^{DS}, k_{on}^{DS} \gg k_{off}^{apt}, [T]k_{on}^{apt}$.

For the first case, where the displacement strand kinetics have been engineered to be much slower than the aptamer kinetics, we have:

$$k_{obs}^{fast} = [T]k_{on}^{apt} + k_{off}^{apt}$$

By applying the binomial approximation to the equation for the slow time constant—which is justified, since $4 * (k_{on}^{DS}k_{off}^{apt} + k_{off}^{DS}(k_{off}^{apt} + [T]k_{on}^{apt})) \ll (k_{off}^{DS} + k_{on}^{DS} + [T]k_{on}^{apt} + k_{off}^{apt})^2$ —we obtain:

$$k_{obs}^{slow} = k_{off}^{DS} + k_{on}^{DS} \frac{k_{off}^{apt}}{[T]k_{on}^{apt} + k_{off}^{apt}}$$

Therefore, in the case where the displacement strand kinetics have been engineered to be slow compared to those of the aptamer, we can arbitrarily decrease the slow time constant. However, the fast time constant is constrained by the native aptamer.

For the second case, where we have engineered the displacement strand kinetics to be much faster than the aptamer binding kinetics:

$$k_{obs}^{fast} = k_{off}^{DS} + k_{on}^{DS}$$

Again, applying the binomial approximation to the equation for the slow rate constant, we find that:

$$k_{obs}^{slow} = k_{off}^{apt} + [T]k_{on}^{apt} \frac{k_{off}^{DS}}{k_{on}^{DS} + k_{off}^{DS}}$$

Therefore, in the case where the displacement strand kinetics have been engineered to be faster than the native aptamer, we can arbitrarily increase the fast time constant, while the slow time constant is constrained by the native aptamer.

The observed kinetics of the ISD construct will always be an additive mixture of two exponential responses based on the slow and fast kinetic rates. The relative magnitude of these contributions is determined, assuming the system begins at equilibrium, by the initial and final target concentrations. From the above two results, we can see that in either limiting design case, one rate constant will be bounded by the native aptamer properties. The deconvolution of fast and slow responses is given by:

$$C(t) = C_{eq} + v_{slow}(v_{slow}^{-1} \cdot C_0)e^{-k_{slow}t} + v_{fast}(v_{fast}^{-1} \cdot C_0)e^{-k_{fast}t}$$

Where the eigenvectors (v) and eigenvalues* are set by the final equilibrium concentrations, and C_0 is determined by the initial concentrations. Depending on the initial and final values of $[T]$, the distribution of k_{fast} versus k_{slow} can be changed via $v_{slow}(v_{slow}^{-1} \cdot C_0)$ and $v_{fast}(v_{fast}^{-1} \cdot C_0)$. Thus, one will always be able to pick some initial and final target concentrations that can induce a binding response that is bounded by the kinetic properties of the native aptamer, regardless of ISD design. However, we note that for slow aptamer kinetics, as long as $v_{fast}(v_{fast}^{-1} \cdot C_0) \gg v_{slow}(v_{slow}^{-1} \cdot C_0)$ and $k_{off}^{DS}, k_{on}^{DS} \gg k_{off}^{apt}, [T]k_{on}^{apt}$, the observed response will be independent of the aptamer kinetics. We have added this description as **Supplemental Note 3**.

*the *mathematica* script with the analytical solutions to the eigenvalues and eigenvectors is provided at <https://github.com/btotherad77/isd>.

3. The reviewer inquired about the practical limits on the lengths of the displacement strand and the loop that can be used to control the switching response.

In our current design (with the displacement strand on the 3' end), the smallest loop length that can be employed is equal to the length of the aptamer minus the length of the displacement strand. Alternatively, the construct could be designed such that the displacement strand originates from the 5' end, in which case the smallest loop length would be ~3 nt. The loop length could be as long as allowed by the constraints of DNA synthesis; for an IDT ultramer, we could reach a maximum loop length of 200 nt - (length of aptamer) - (length of DS). We note that increasing the loop length has diminishing returns (**equation S5**); as the entire loop is lengthened, each additional base has a smaller impact on both the thermodynamics and kinetics. The displacement strand could be as long as the aptamer itself, or longer if bulges were included. However, such lengths would result in a very high K_D^{eff} . And although the displacement strand could be as short as 2 nt in theory, we have observed that a 5 nt displacement strand produces very high background signal, and we wouldn't suggest going any shorter than that without incorporating modified bases such as peptide nucleic acids (PNA) or locked nucleic acids (LNA).

REVIEWERS' COMMENTS:

Reviewer #1 (Remarks to the Author):

Wilson et al. provided a detailed response to the reviewer's comments and conducted additional experiments that support their conclusions. As such, we are pleased to recommend publication of the manuscript.

Jeffrey Munzar, Andy Ng and David Juncker

Reviewer #2 (Remarks to the Author):

As I stated in my previous evaluation, the manuscript presents a general strategy for transforming aptamers into molecular switches, analysing the design instructions for controlling in an independent way affinity and kinetics of the recognition event. After revision, authors included new experimental data, and theoretical analysis, which demonstrate the usefulness of the approach as well as its limitations. Thus, I recommend publication in Nature Communications.